# Challenges in Serologic Diagnostics of Neglected Human Systemic Mycoses: An Overview on Characterization of New Targets

**DOI:** 10.3390/pathogens11050569

**Published:** 2022-05-11

**Authors:** Moisés Morais Inácio, Vanessa Rafaela Milhomem Cruz-Leite, André Luís Elias Moreira, Karine Mattos, Juliano Domiraci Paccez, Orville Hernandez Ruiz, James Venturini, Marcia de Souza Carvalho Melhem, Anamaria Mello Miranda Paniago, Célia Maria de Almeida Soares, Simone Schneider Weber, Clayton Luiz Borges

**Affiliations:** 1Laboratory of Molecular Biology, Institute of Biological Sciences, Federal University of Goiás, Goiânia 74690-900, Brazil; moises.biomed@gmail.com (M.M.I.); van-rafaela@hotmail.com (V.R.M.C.-L.); andre.bio.br@hotmail.com (A.L.E.M.); julianopaccez@gmail.com (J.D.P.); cmasoares@gmail.com (C.M.d.A.S.); 2Estácio de Goiás University Center, Goiânia 74063-010, Brazil; 3Faculty of Medicine, Federal University of Mato Grosso do Sul, Campo Grande 79070-900, Brazil; karinee.mattos@gmail.com (K.M.); ja-mes.venturini@ufms.br (J.V.); marcia.melhemufms@gmail.com (M.d.S.C.M.); 4MICROBA Research Group—Cellular and Molecular Biology Unit—CIB, School of Microbiology, University of Antioquia, Medellín 050010, Colombia; orville.hernandez@udea.edu.co; 5Bioscience Laboratory, Faculty of Pharmaceutical Sciences, Food and Nutrition, Federal University of Mato Grosso do Sul, Campo Grande 79070-900, Brazil; anapaniago@yahoo.com.br

**Keywords:** systemic mycoses, diagnosis, new targets, in silico approaches, experimental approaches

## Abstract

Systemic mycoses have been viewed as neglected diseases and they are responsible for deaths and disabilities around the world. Rapid, low-cost, simple, highly-specific and sensitive diagnostic tests are critical components of patient care, disease control and active surveillance. However, the diagnosis of fungal infections represents a great challenge because of the decline in the expertise needed for identifying fungi, and a reduced number of instruments and assays specific to fungal identification. Unfortunately, time of diagnosis is one of the most important risk factors for mortality rates from many of the systemic mycoses. In addition, phenotypic and biochemical identification methods are often time-consuming, which has created an increasing demand for new methods of fungal identification. In this review, we discuss the current context of the diagnosis of the main systemic mycoses and propose alternative approaches for the identification of new targets for fungal pathogens, which can help in the development of new diagnostic tests.

## 1. Human Systemic Mycosis

Fungal infections are a problem faced by developed and developing countries. Worldwide, diseases caused by pathogenic fungi have led to various socio-economic consequences and physical sequelae which can result in the death of many individuals. In fact, over a billion people, immunocompetent and immunocompromised individuals, are affected globally by fungal diseases [1]. The rate of dissemination and infection is a consequence of the ability to overcome the immune system, tolerate host temperature and uptake micro- and macro-nutrients [2].

In this review, we will focus our discussion on the serologic diagnostics of a group of neglected human systemic mycoses caused by fungi, such as *Paracoccidioides* spp., *Histoplasma capsulatum*, *Aspergillus* spp., *Coccidioides* spp., as well as *Cryptococcus* spp., which are responsible for paracoccidioidomycosis (PCM), histoplasmosis, aspergillosis, coccidioidomycosis and cryptococcosis, respectively. These mycoses commonly present as a pulmonary disease, and disseminate into tissues and systems, thus affecting the work capacity of patients and having a negative impact on the health systems of countries where they prevail. Biological knowledge about these fungi is key for the development of strategies to combat them and may represent a starting point for rapid diagnostic tests, as well as therapy and vaccine development. 

## 2. Neglected Human Systemic Mycoses Diagnosis

The diagnosis of systemic mycoses can often be a challenge; however, it is very important to ensure the most appropriate treatment and clinical follow-up to monitor treatment effectiveness and side effects [1]. In this sense, a fast and accurate diagnosis could reduce the empirical antifungal therapies, impact on evolutionary selection pressure and contribute to resistance emergence management [2]. The main challenges are the lack of sensitive and specific methods for early diagnosis, the lack of standardization of serological and molecular tests, the wide antigenic variability of the clinical isolates, and the fastidious and slow-growing nature of some fungal species [1]. In addition, in low-prevalence areas, the positive predictive values of some nonculture-based tests could be significantly lower than in endemic areas.

Overall, histopathologic, direct and culture examinations from clinical samples are often used as the standard diagnostic for systemic mycoses. Although attempts to culture the microorganisms should always be pursued, culture is less effective when the fungal burden is low or depending on the clinical form or type of fungal infection. For instance, *Pneumocystis jirovecii* do not grow in vitro [3], and *Paracoccidioides* species are usually isolated in only 10 to 20% of culture examinations and can take up to a month to grow. Detection of antibodies or antigens provides valuable information about current disease and is important for the management of fungal infections. However, it is often unavailable for most mycoses. In addition, molecular approaches could be useful in detecting fungal DNA in low fungal burden cases, mainly from biological samples, but these approaches are still not well standardized (Table 1). Despite the challenges above, efforts to properly identify the pathological agent are pivotal, since early treatment, which depends on the correct diagnosis, can prevent complications and help to reduce the morbidity and mortality of the systemic fungal infections.

### 2.1. Paracoccidioidomycosis

*Paracoccidioides* spp. are endemic fungi restricted to Latin America [3,4,5]. The genus is composed of six human pathogenic species, *Paracoccidioides brasiliensis*, *P. lutzii*, *P. americana*, *P. restrepiensis*, *P. venezuelensis* [6] and the non-culturable *P. loboi* [7]. Brazil has a high incidence of PCM in the South, Southeast and Midwest regions, where the prevalence is related to agricultural work [5,8,9]. Rural workers are the most affected individuals, where males have a greater PCM distribution compared to females, which can be explained by female hormones [10]. The infection is triggered by the inhalation of conidia or mycelia propagules [5,11]. Inside the host, the inhaled fungus produces the yeast form that can disseminate to several sites, causing the acute/subacute or chronic disease forms, depending on host-parasite interaction [10,12].

The gold standard for definitively diagnosing PCM is the visualization of yeast cells with typical multiple budding aspects (“ship-pilot´s wheel” or “Mickey Mouse head”) in fresh examination of sputum or other clinical samples (scraped from lesion, lymph node aspiration, lesion sample, etc.). However, a low percentage of positive samples result in positive cultures. The detection of serum antibodies is diagnostically valuable and is particularly important to clinical follow-up as it allows the evaluation of the host response during specific antifungal treatment. The DID test has high specificity (100%) and sensitivity (from 65 to 100%) depending on the fungus isolate, endemic area of isolation, the culture conditions, the technique used and antigen profile [11,12,13,14]. In a comparison between WB versus DID with a purified glycoprotein of 43,000 Daltons from *P. brasiliensis* (Gp43), DID showed sensitivity (80%) and specificity (>90) of the test in an endemic area (different regions of Brazil), without false positive results or cross reaction [14]. Although ELISA (88–95% of specificity) is less specific than DID, mainly due to cross-reaction with histoplasmosis [15], it is highly sensitive (up to 100%), fast and suitable for PCM high-throughput screening [16]. Overall, among available serological techniques in the detection of serum antibodies, DID is the best method in patients with suspected PCM. On the other hand, Gp43 and Gp70 were described as good markers for monitoring antigen clearance during antifungal treatment by ELISA assay [17,18]. The usage of the Gp43 marker has become less useful since this antigen is usually not identified in the infections caused by *P. lutzii* [16,19]. The most common specimen employed in serological diagnostics is patient sera; however, the use of cerebrospinal fluid and bronchoalveolar lavage (BAL) specimens increases the sensitivity for antigen detection in the central nervous system and in pulmonary infections, respectively [20].

**Table 1 pathogens-11-00569-t001:** Neglected Human Systemic Mycoses Diagnosis.

Mycosis	Diagnostic Test	Human Specimen	Time until Results	Accuracy	Advantage	Disadvantage	Infrastructural Resources	References
Paracoccidioidomycosis (PCM)	Double Immunodiffusion test	Serum	This technique requires much time.	Specificity 100% and Sensitivity 65–100%	The choice method in patients with suspected PCM without false positive results or cross reaction.	Low accuracy for determination of the patient’s cellular immunity important during the therapy.	Laborious. However, it is not expensive.	[12,13,17]
	ELISA	Serum	This technique is fast.	Specificity 88–95%. Sensitivity 100%	Fast and suitable for PCM high-throughput screening.	Cross-reaction with Histoplasmmosis	Requires specific equipment and automatization.	[15,16]
	Nucleic Acid testing	Blood, and clinical specimen	Result is obtained in a few hours.	Specificity 100%	Genotypic studies and clinical diagnosis performed directly from samples.	Need to standardize techniques based on DNA amplification for its real implementation.	Requires a specific and high-cost equipment.	[21]
Histoplasmosis	Immunodiffusion	Serum	This technique requires much time.	Sensitivity 60–70%	Rapid turnaround time	Not be used in immunocompromised individuals, since this group may present increases in false-negative results due to the compromise of the humoral response.	Low costs and simple infrastructure	[22]
	Complement Fixation	Serum	This technique requires much time.	Sensitivity 60–70%	Rapid turnaround time		Laborious technique and requires well trained personnel	[22]
	Enzyme Immunoassay	Serum, Plasma, Urine/CSF/BAL/Other Body Fluid.	Fast. (approximately 1.5 hours)	Sensitivity ranges 95–100% in urine, over 90% in serum and BAL antigens and 78% in cerebral spinal fluid (CSF)	Particularly important in AIDS patients who have disseminated histoplasmosis and who have large fungal burden	Serologic cross-reactions to Histoplasma-like antigens with Blastomycosis, Coccidioidomycosis, PCM and Aspergillosis	Requirement of specialized laboratories, expensive equipment, and well-trained personnel	[22,23]
	Nucleic Acid testing	Blood and other body fluid.	The results are obtained in a few hours.	Specificity 100% and a sensitivity 67% to 100%.	Genotypic studies and clinical diagnosis performed directly from samples	Need to standardize techniques based on DNA amplification for its real implementation.	Requires specific and high-cost equipment.	[24,25]
Aspergillosis	ELISA assays galactomannan (GM) detection *	Serum, lung transplant recipients, sputum or bronchoalveolar lavage.	This technique is fast.	Sensitivity 60 to 100% and specificity 85 to 98%	GM levels are proportional to fungal burden in tissue, and present prognostic value	Both false positive and false negative results have been reported and cross reactivity.	Performed without the need for specialized equipment and reagents	[26,27,28]
	Nucleic Acid testing	Serum, Lung transplant recipients, sputum or bronchoalveolar lavage.	The results are obtained in a few hours.	Specificity 100%	More sensitive and quick diagnosis	The lack of sensitivity and the difficulty in distinguishing between infection and colonization.	Requires specific equipment and has a high cost.	[29,30,31].
Coccidioidomycosis	Direct examination	Sputum or bronchoalveolar lavage or other biopsy material	This technique is fast.	N/A	The gold standard diagnostic method	The mold form of *Coccidioides* produces highly infectious arthroconidia as soon as 72 hours after initial growth.	Requires well trained personnel	[30,32]
	Culture	Sputum or bronchoalveolar lavage or other biopsy material	Requires a lot of time.	N/A	The gold standard diagnostic method	This form represents a significant risk of inhalational exposure to laboratory personnel.	The potential exposure risks associated with aerosolization	[30,32,33]
	Enzyme immunoassays	Serum, urinary, and cerebrospinal fluid	This technique is fast.	Sensitivity 88%. Specificity 90%	Antibody detection EIA is a sensitive and specific test, including high-risk patients’ samples, in detection of IgG and IgM antibodies	Maybe insensitive to early infection.	Performed without the need for specialized equipment and reagents	[30,32,34]
Cryptococcosis	Direct examination	Biopsy material	This technique is fast.	Sensitivity 60–90%	More sensitive and quick diagnosis	Lower sensitivity in HIV-negative patients in association with a low fungal burden.	Low-resource method	[35,36]
	Culture exam	cerebrospinal fluid	1 to 2 weeks for definitive results	Sensitivity 85–95%	More sensitive. A gold standard for diagnostic	Need longer incubation periods up to three weeks.	The cultures are easily performed in any microbiology laboratory.	[37]
	Nucleic Acid testing	Plasma or cerebrospinal fluid	The results are obtained in a few hours.	Specificity 100%	Allows the determination of the *Cryptococcus* species	Need to standardize techniques based on DNA amplification for its real implementation.	Requires a specific and high-cost equipment.	[38]
	lateral flow assay	Plasma or cerebrospinal fluid	This technique is fast.	Sensitivity 90–100%	Provides a rapid diagnosis of cryptococcosis by detecting capsular antigen of *Cryptococcus* spp. In serum, plasma or CSF.	Low specificity (false positive 11% to 14%)	Low-costs	[39,40]

* Unusual in routine or applied in specific situations.

It is known that PCM control depends on an effective cellular immune response [41]. In this context, in order to prevent disease recurrence, it has been suggested that the suspension of treatment which is, in fact, too prolonged should occur after the patient’s cellular immune response recovery. So far, there is no specific test to assess the cellular immune response of PCM patients during treatment. Even though DID only assesses humoral immunity [17], the serological cure has often been used as an immunological cure parameter. However, this correlation is not accurate, since the DID assay has little antibody-detection power, and its low accuracy prevents the method’s validation for determining the patient’s cellular immunity. Depending on the host’s immune response and/or serological tests, patients may present high antibody titers at the end of treatment without presenting disease symptoms, while others may have low titers even in the presence of clinical symptoms [42].

Another factor that limits the use of the DID assay as a patient follow-up is the fact that about 10 to 40% of patients with PCM may not show positivity in the immunodiffusion assay [6]. This is probably due to the different antigenic profiles of the *Paracoccidioides* species and may affect the diagnosis of PCM. In this sense, our group recently identified a set of B-cell epitopes exclusive to the *Paracoccidioides* complex and a set specific to each fungal species, which were developed from an immunoproteomic approach [43]. These epitopes demonstrated promising results on serological tests (data not shown), however, they still need to be widely validated. 

### 2.2. Histoplasmosis

*Histoplasma capsulatum* is the causative agent of American histoplasmosis in both immunocompromised and immunocompetent individuals. The mycosis is the largely distributed in North America; however, it has broken through the barriers of the endemic areas of Ohio and the Mississippi River and is found in other regions around the world [44,45,46]. The presence of microconidia or small hyphae fragments of *H. capsulatum* in soil contaminated by bird or bat feces is the primary scenario of host-contact, where the infection occurs via inhalation of the airborne fragments [47,48]. The histoplasmosis can manifest as acute, subacute, and chronic pulmonary in immunocompetent individuals, while the disseminated cases are more common in immunocompromised individuals, particularly during HIV infection [49,50].

Early and rapid detection of histoplasmosis is essential in preventing morbidity and mortality, but remains challenging mainly in impaired immune system patients, such as individuals suffering from AIDS. In immunocompromised individuals, histoplasmosis becomes progressive and spreads rapidly from the lungs to other organs and is known as progressive disseminated histoplasmosis (PDH) [25]. The definitive diagnosis of histoplasmosis is accomplished by isolation of *H. capsulatum* in culture, as well as by visualization of the yeast form in samples [22]. However, these procedures lack sensitivity and are time consuming. Thus, antibody detection methods represent the major tools currently in use for non-culture diagnosis, predominantly because of their availability and rapid turnaround time. Nevertheless, this method should not be used in immunocompromised individuals, since this group may present an increase in false-negative results due to the compromised humoral response. Furthermore, serologic cross-reactions to Histoplasma-like antigens occur in patients presenting other systemic mycoses, such as Blastomycosis, Coccidioidomycosis, PCM and Aspergillosis [51,52,53]. 

Due to the high specificity of *H. capsulatum*, the identification of anti-H and anti-M antibodies using antigenic extract (histoplasmin) from mycelial culture is notably useful in serological diagnosis of histoplasmosis. The sensitivity of antibody detection by ID or CF is between 60% and 70% [22]. On the other hand, detection of the circulating *H. capsulatum* polysaccharide antigen in urine and serum is particularly important in AIDS patients who have disseminated histoplasmosis. Antigenuria can be used for monitoring the host response to antifungal treatment [22], while identification of antigens in BAL is useful in pulmonary histoplasmosis [54]. Recent advances highlight MiraVista Diagnostics, a company that developed three generations of EIA assay, with sensitivity ranges of between 95–100% in urine, over 90% in serum and BAL antigens, and 78% in cerebral spinal fluid (CSF) [22]. However, the high cost of testing is still an obstacle in using the assay for the diagnosis of histoplasmosis. 

### 2.3. Coccidioidomycosis

Another endemic mycosis that primarily affects the lungs is coccidioidomycosis, caused by *Coccidioides* spp. This genus comprises *Coccidioides immitis* and *Coccidioides posadasii*, which also cause pulmonary disease in immunocompetent individuals. The endemic region of *Coccidioides* spp. is that with an arid climate where the infection occurs predominantly in the dry seasons [55]. *Coccidioides* spp. are prevalent in Mexico and the southwestern United States (Arizona, Texas and California) where the endemic areas were determined by skin test using spherulin or coccidioidin antigen preparations [56,57]. In South America, *Coccidioides* is present in Bolivia, Paraguay, Argentina, and Brazil [58,59,60]. The infection occurs after fungi arthroconidia inhalation, which reach the host’s pulmonary system and undergo dimorphic transition to yeast or spherule infective forms [57,61]. The clinical manifestation of coccidioidomycosis ranges from pulmonary infection to life-threatening pneumonia, to the dissemination of the infection to the tissues of the human body [62]. 

Clinical information is important, but the identification of *Coccidioides* on pathologic examinations or the isolation of fungus in culture are the gold standards for diagnosis of coccidioidomycosis. However, the use of these techniques is less frequent compared to the large number of actual cases, lack sensitivity, are time consuming, and require a degree of expertise to recognize the fungus [31,33,63].

Serology is the most used method of diagnosis [64]. Among methods used, EIA for antibody detection is the most regularly employed. Furthermore, two methods are commercially available, the Meridian Premier *Coccidioides* EIA (Cincinnati, OH, USA) and the Immuno-Mycologics Inc. (IMMY) Omega *Coccidioides* EIA (Norman, OK, USA), which are performed in most major reference laboratories and some laboratory hospitals. An alternative test is performed by detecting immunoglobulin G (IgG) by CF or ID and immunoglobulin M (IgM) antibodies by immunodiffusion. However, EIA is simpler to perform and provides same-day results, while ID and CF are difficult to execute and require 2 to 6 days to provide results [65]. 

A new test was developed by MiraVista Diagnostic (MVista), which demonstrates a sensitivity for IgG and/or IgM of 88% compared to 60% for ID and 66% for CF. Furthermore, the EIA MVista maintained similarly high sensitivity in immunocompromised patients (IgG 83% and IgM 56%) for whom ID sensitivity was reduced (IgG 40% and IgM 30%). It also maintained a 90% specificity and demonstrated low-to-moderate rates of cross-reactivity with other endemic mycoses (32% histoplasmosis and 8% blastomycosis) [34].

Malo et al. (2020) evaluated three commercial enzyme immunoassay kits: the IMMY omega EIA and the Meridian Premier EIA (for IgG detection) and IgM with the new EIA test, MVista Coccidioides test, and observed that the sensitivity of the IgG antibody detection was 87.4% using the MVista test compared to 46.6% for the IMMY test and 70.9% for the Meridian test. Similarly, the specificity of IgG and IgM antibodies was higher for the MVista EIA (90% and 95.3%, respectively), indicating that the MVista Coccidioides antibody detection EIA is a sensitive and specific test, including high-risk patients’ samples, in the detection of IgG and IgM antibodies [65].

Nevertheless, serologic tests for coccidioidomycosis may be insensitive to early infection [66]. Therefore, to minimize the potential for false-negative testing, serial serological testing is recommended, necessitating an initial serology followed by a second specimen from the convalescent phase of the disease. On the other hand, an isolated positive IgM EIA test should be followed closely with clinical correlation and subsequent diagnostic testing [66]. In many instances, this will generate serial testing, repeat EIA testing for IgM and IgG, or confirmatory testing by the immunodiffusion-tube precipitin reaction [67].

Recently, a monoclonal antibody ELISA test against coccidioidal CTS1 antigen has been developed [68]. CTS1 is also known as the “CF” antigen also used in coccidioidomycosis serodiagnosis by CF and ID tests.

### 2.4. Aspergillosis

*Aspergillus* species are agents of pulmonary aspergillosis. Despite numbering in their hundreds, only a few species have an impact on human health. The most clinically important fungi are: *Aspergillus fumigatus* [69,70], *A. flavus* [71] *A. nidulans* [72], *A. niger* [73] and *A. terreus* [74]. Aspergillus-related pathologies are caused by inhalation of airborne conidia which are encountered in air, soil, water, outdoor plants, as well as in hospitals [75]. *Aspergillus* species continue to be an important cause of life-threatening infection in immunocompromised patients, particularly those under prolonged corticosteroid therapy, immunosuppressive drugs, or with hematological malignancies, or people infected with HIV or individuals suffering from AIDS [76]. Aspergillosis presents with a wide range of clinical syndromes, including allergic bronchopulmonary aspergillosis (ABPA), chronic pulmonary aspergillosis (CPA), and the most severe, invasive pulmonary aspergillosis (IPA), which is linked to high mortality rates [70,75,77] Chronic pulmonary aspergillosis has been recognized as an important and neglected fungal infection [78,79,80,81]. It is estimated that approximately three million cases of CPA aspergillosis occurred annually, being 2000 to 160,000 people after tuberculosis treatment, annually, with 50% case fatality in 5 years [82].

Diagnosis of aspergilloma or invasive aspergillosis (IA) can be difficult, and currently alternatives include laboratory tests such as histopathologic/cytologic and culture examination. Serum detection of galactomannan (GM) and (1,3)-beta-D glucan (BDG) were also recommended to be used in patients with specific clinical conditions, such as hematologic malignancy as well as submitted to allogeneic hematopoietic stem cell transplant [83,84]. GM is a cell wall component of *Aspergillus* that is released by the fungus [26] and can be detected through latex agglutination or ELISA assays. It has been suggested that GM levels are proportional to the fungal burden in tissues and present prognostic value [27]. In this context, IMMY diagnostics has developed an antigen assay lateral flow device with a sensitivity of 40% and specificity of 80%. In addition, the GM assay uses sandwich *EIA* for the diagnosis of invasive aspergillosis, although it can also be found on the *H. capsulatum* and *Fusarium* spp. cell walls [85,86].

CPA is possibly the systemic mycosis where serological testing is most useful. While antigen detection tests such as galactomannan are important in IA, it is only positive in about 25% of CPA patients [87]. Anti-aspergillus tests are, however, positive in over 90% of patients. In practice, precipitation techniques were replaced by an ELISA IgG antibody detection test [88], because it was the fastest, the most sensitive [89], and easily automated, although it is lacking in specificity. In the last decade, several commercial Aspergillus-specific IgG ELISAs have been developed and are now widely used in CPA diagnosis [88]. 

Finally, the ABPA is a hypersensitivity reaction test to *Aspergillus* species (generally *A. fumigatus*) that occurs almost exclusively in patients with cystic fibrosis or, less commonly, with asthma. The ABPA diagnosis can be confirmed by IgE levels and *A. fumigatus*–specific antibody detection and by visualization of the yeast forms in clinical samples with the help of fungus-enhancing staining, such as calcofluor white, or using histopathology techniques [84].

### 2.5. Cryptococcosis

Cryptococcal meningitis, the most severe form of this systemic mycosis, it is listed in the G-Finder report, and meets the WHO and PLOS definitions of a neglected disease, affecting disproportionately populations in poverty, causing high morbidity and mortality, and being neglected by research [81,90,91].

Two complexes of the *Cryptococcus* species, which determine epidemiologically distinct clinical conditions, are responsible for cryptococcosis. *Cryptococcus neoformans* and *C. gattii* are associated with conditions of host cellular immunodepression. In the past, *C. gattii*, was considered the agent of primary cryptococcosis in apparently normal hosts, however, both have been found in immunocompromised and immunocompetent individuals [92,93]. Furthermore, both complexes cause meningoencephalitis, which has a severe and fatal evolution and can be accompanied by evident lung injury. Additional effects of the infection include fungemia and secondary foci for skin, bones, kidneys, and adrenal, among others [93,94].

Currently, the diagnosis of cryptococcosis, as well as most systemic mycoses, is performed in routine laboratories by mycological examination using histopathological and immunological tests. For better observation of the capsule, a drop of Indian ink dye should be added on a microscope slide. The sensitivity of this test assessed on CSF samples ranges from 60% to 90%, according to the analyst’s expertise and according to the fungal load. However, direct examination of blood samples has low sensitivity, and its use has not been recommended [35,36,95]. 

Isolation of the etiologic agent by culture allows morphological, biochemical and molecular analysis of the clinical strain. For CSF samples, the culture exam sensitivity is from 85% to 95%, according to the disease stage and fungal load [37]. The presumptive identification of the *Cryptococcus* genus is performed by microscopic examination of a portion of the culture that allows the presence of yeast to be checked with a capsule without the production of hyphae (some strains may form short pseudo hyphae). The positive urease test minimally complements the presumptive genus identification. The characteristic production of melanin in *Guizzotia absynica* agar is essential in the diagnostic laboratory, to show that it is a question of which pathogenic species is being examined [38]. For histopathological examination, the sample should be stained with Mayer’s mucicarmine, that highlights the capsule in red and is therefore particularly useful in hypocapsulated strain infections [96]. Meanwhile, the Fontana–Masson staining shows the melanin of the cell wall and, therefore, it is also specific for both *Cryptococcus* complexes species, because in the other *Cryptococcus* members the reaction is negative. Other methods, such as periodic acid from Schiff (PAS) and Groccot–Gomori can be used, however, they are non-specific. Hematoxylin–eosin staining is also useful to check the tissue reaction profile, which is quite poor in general, but eventually allows the verification of granulomatous reaction in immunocompetent patients or infections with hypocapsulated strains [97]. In addition, the MALDI-TOF MS approach allows the correct identification of species from the two complexes, provided commercial standard databases are properly enriched [98,99].

Immunological examination provides a rapid diagnosis of cryptococcosis by detecting the capsular antigen of Cryptococcus spp. in serum, plasma or CSF. Quantitative examination by antigen titration has a prognostic value. In cryptococcal infection, unlike other invasive mycoses, the humoral response, assessed by antibody quantification, is poor and therefore this method is not suitable for diagnosis. Otherwise, the capsular antigen (*Cryptococcus* Antigen-CrAg), represented by polysaccharide molecules, is soluble in several body fluids and can be detected in CSF in cases of meningitis and in the serum of patients with and without meningitis weeks to months before symptoms of the disease present [100]. All species of *Cryptococcus* spp. have capsular antigens and, therefore, a positive test indicates active disease, but does not allow identification of the species of the etiologic agent.

The search for capsular antigens was traditionally carried out by an agglutination test with latex particles sensitized with antibodies against *Cryptococcus*. A more recent qualitative and quantitative test for capsular antigens carried out on strip has been available. It was developed to be cheaper and simpler, with the point of care based on the immunochromatography reaction denominated lateral flow assay (LFA). The test stripes can be stored, for up to two years, outside refrigeration and have been designated as point-of-care tests. Other manufacturers, from France, China, and Denmark, commercialize LFA for *cryptococcosis* diagnosis, but few studies have been conducted for evaluating the performance of the new trend marks, in comparison to the IMMY LFA. The majority of the following information was obtained using the north American test [101]. The LFA test is more sensitive in the detection of polysaccharides from the four *Cryptococcus* serotypes compared to the latex test. For children, data on the accuracy of the test are limited, although it is estimated that its performance is similar to that observed in adults [102].

The LFA test has greater analytical sensitivity compared to others when it comes to antigens of both *C. neoformans* and *C. gattii* [103]. A small amount of sample is sufficient to diagnose cryptococcal meningitis, initial or advanced, with sensitivity from 97.6% to 100% (serum) and between 94.0% to 100% (CSF) [40,89,104,105] without the need for laboratory equipment or technical expertise, which makes the test ideal for regions with limited resources [106]. The specificity of the test in serum reaches 98.1% and in CSF 98.9%. For pulmonary forms there are few studies with LFA, indicating that the test is better than latex, since the sensitivity reaches 100% with serum samples [102].

For HIV-positive patients with cryptococcal meningitis, samples of whole blood, collected on a fingertip, eluted on the LFA test strip, resulted in equal sensitivity (95%) to that obtained with serum and CSF samples. The specificity of the test, performed with sera from HIV positive patients with other diseases of the central nervous system, was 100%, but urine samples from these patients were positive in only 80% of the cases in the same study, indicating limitation in the use of this sample for antigen research [107]. In fact, the specificity of the test with urine samples is still too unreliable to recommend this procedure [108,109].

LFA is particularly useful in patients with HIV infection in whom early detection of cryptococcosis, followed by pre-emptive antifungal therapy with fluconazole (screen and treat strategy), reduces disease progress and mortality from meningitis [109,110,111,112]. The WHO recommends antigen research (serum, CSF, plasma, whole blood) for screening cryptococcosis in adult and adolescent populations in regions where the prevalence of cryptococcal antigenemia is high (>3%), although it appears to be cost-effective in regions with lower antigenic prevalence, such as 0.06% [39,113]. 

## 3. Approaches Used to Search for New Diagnostic Candidates

### 3.1. Experimental Approaches

#### 3.1.1. Cell-Free Antigens and Total Exoantigens

In the past, techniques used to obtain secreted antigens were called Cell-Free Antigens (CFAg) [114,115,116]. Currently, a new nomenclature called exoantigens (ExoAg) is used for these secreted antigenic molecules [43,117,118]. This technique consists of obtaining molecules secreted by pathogenic organisms. For execution of the technique, the initial cultivation of fungi in semi-solid culture medium is necessary, followed by inoculation in liquid culture medium, under constant temperatures (25–37 °C) and agitation, remaining for 48 hours, or up to 14 days depending on the microorganisms [43,114,119,120,121,122]. During this period, the fungus secretes molecules into the extracellular environment. Subsequently, the culture supernatant is collected and purified by means of filtration membranes in order to separate the cells and obtain only the molecules secreted by the organism [43,123,124].

Next, CFAg can be used for various purposes. It might be used in immunological diagnostics, where techniques such as immunoblotting are employed [125,126,127], ELISA as well as ID [126,127]. Immunoproteomic techniques can also be applied to identify new antigenic targets [43,128]. In addition, exoantigens can be used in studies focused on cellular immunization processes [115,129,130] (Table 2).

One of the first reports on the use of ExoAg from fungi that cause systemic mycoses was performed by Standard and Kaufman [131], where immunological assays were used for specific and rapid detection of ExoAg from *H. capsulatum*. In other studies, exoantigens from *Sporothrix schenckii* were used for the rapid and specific identification of this organism in mycelial cultures [132]. In 1990, this technique was performed using exoantigens from *P. brasiliensis* and serum from sick patients, with the objective of performing the immunodiagnosis of PCM. This study resulted in the identification of GP43, one of the main antigens identified in *Paracoccidioides* sp. [114].

In studies performed by Sá-Nunes et al., [115] mice infected with *H. capsulatum* were immunized with ExoAg, resulting in the immunization of mice against histoplasmosis. Furthermore, this immunization proved to be more efficient compared to immunizations performed with histoplasmin antigen [115]. In other works, aimed at developing and characterizing a new approach to immunization against histoplasmosis, microspheres were used that would act in the controlled release of secreted antigens from *H. capsulatum* [116]. These assays showed the production of pro-inflammatory cytokines during in vitro tests, presented the potential to be used as vaccines, and to provide protection against *Histoplasma* sp. [116]. In other studies, Culture Filtrate Protein 4 (Cfp4) of *H. capsulatum* was identified as secreted and showed an important role in the pathogenesis of the fungus. In addition, it has been successfully probed by anti-Cfp4 monoclonal antibodies, showing the importance of this exoantigen as a possible molecule to be used in the diagnosis of histoplasmosis [118]. Moreover, the M and H antigen had their biological characteristics described [133,134,135,136], and these molecules were initially identified on the yeast cell surface of *H. capsulatum*. However, recent studies have shown the secretion of these molecules into the extracellular environment and their antigenic potential, making them useful antigens for the diagnosis of histoplasmosis [128].

Extracellular vesicles carrying highly immunogenic epitopes have already been described in *P. brasiliensis* [125]. In these vesicles, antigens of the fungus were found that are present in different phylogenetic isolates. Additionally, epitopes linked to α-galactopyranosyl were found in exosomes, showing that they are highly immunogenic, as evidenced by immunoblotting and ELISA [125].

**Table 2 pathogens-11-00569-t002:** Recommended approaches to search for new diagnostic candidates.

Experimental Approaches	Advantage	Disadvantage	Infrastructure	References
Cell-free antigens and total exoantigens	Can be used for various purposes, such as immunological diagnostic techniques and to identify new antigenic targets and studies of cellular immunization processes.	The sensitivity and specificity of the tests are related to the production of the antigen, have high possibility of cross-reaction and difficulty in diagnosis at the beginning of the disease.	Low-cost technique and simple to perform. Requires laboratory infrastructure for incubation of microorganism and purification of secreted molecules.	[124,129,130,137,138,139]
Immunoproteomics	Has been used successfully for the identification and characterization of antigens applied as new markers for molecular diagnostics, as well as possible candidates for vaccine production used in therapies.	This technique requires antibodies with high selectivity or sensibility and capture of specific antigens in crude samples	Requires sophisticated laboratory infrastructure, trained professionals, and high cost.	[43,140,141,142]
Peptide microarrays	Provides an extremely rapid and robust method which allows thousands of targets to be tested simultaneously.	This approach is not sufficient to cover complete proteomes. Furthermore, it is difficult to bind antibodies that need specific conformations and longer sequences.	It is not a high-cost technique, but it requires infrastructure and specialized professionals.	[143,144]
Cell surface shaving	Effective in identifying antigenic proteins exposed on the cell surface, which proves to be one of the best targets for host immunity.	This technique is less used in Gram-negative microorganisms, due to the thinner cell wall that does not resist digestion without lysis.	High-cost technique, which requires specialized laboratory and trained professionals.	[145,146]
Phage display	Allows rapid identification and isolation of highly specific phage.	Need of phage display libraries construction with stability, quality, and diversity of antibody. Furthermore, difficult to select antibodies against the antigens which are expressed on the surface of rare cells.	Low-cost technique and simple to perform.	[147,148,149]
Bioinformatics analysis	It is possible to map a specific antigen and to identify the epitope with great potential for targets in vaccine and diagnosis development. In silico analyses are faster and more cost-effective, and the possibility of identifying proteins that are not expressed in vitro.	The target identified by in silico analysis need experimental confirmation.	Low-resource method. The infrastructure consists of a computer, internet and trained personnel.	[150,151]

Other works aimed at the identification and application of these molecules in the PCM serodiagnosis have been carried out. Studies, such as the one by Perenha-Viana et al. [14], analyzed 517 patient sera using WB and DID, using ExoAg from *P. brasiliensis*. All tested sera showed positive reactivity through DID and showed positive results from GP43 through WB. Interestingly, GP43 has also been identified as an immunodominant ExoAg for the chronic form of PCM [122]. In addition, in studies using ExoAg GP43, it was possible to verify a great antigenic variability in the studied geographic regions. Moreover, differences in immunoreactive bands were observed between isolates of *P. lutzii* (*Pl*2875, *Pl*9840 and *Pl*2912) and *P. brasiliensis* (*Pb*166 and *Pb*2880) [152].

#### 3.1.2. Immunoproteomics

With the advances in studies and the development of several proteomic and immunological techniques, a new area called immunoproteomics has arisen [142,153]. Recently, this approach has been used successfully for the identification and characterization of antigens that can be applied as new markers for molecular diagnostics, as well as possible candidates for vaccine production for use in therapies [128,139,140,154,155,156]. In this sense, this new approach is used to describe molecular techniques that make it possible to identify antigens on a large scale [142].

In order to perform immunoproteomic analyses, circulating antibodies from an affected host are used to bind to the pathogen-specific antigens, followed by antigen isolation and identification by MS [157]. These techniques have already been used for studies of fungal antigens, such as two-dimensional gel electrophoresis, immunoblotting, polyclonal antibody production, LC MS/MS and MALDI-TOF MS, ID, immunoprecipitation, cytokine assays and bioinformatics [42,43,153,154,158,159]. Some techniques, such as immunoblotting coupled to mass spectrometry, are of great importance for carrying out immunoproteomic analysis. Immunoblotting is an approach that uses antibodies from an affected individual to probe immobilized antigens on a surface. This approach is highly sensitive, allowing the recognition of poorly concentrated antigens in analyzed samples. Thus, this technique can be used both in immunological diagnosis and in scientific research, aiming at the identification and characterization of new antigens [160].

For the identification of antigens, the high-performance liquid chromatography-mass spectrometry (HPLC-MS) stands out with excellent results. This tool provides the most robust analyses, where a complex of samples can be prepared in a single tube and all the processes associated with chromatography and MS and MS/MS and data processing can be performed in a few steps [161,162].

Thus, this type of analysis has been used successfully to identify biomarkers for fungal diseases, including candidiasis [156,163,164], paracoccidioidomycosis [43,140], cryptococcosis [158,165], and is also aiming at the development of vaccines, diagnostic tests [154,166], clinical biomarkers [155,167] and comparative analyses [159,168]. Due to their effectiveness in the identification of antigenic molecules, these techniques have already been used in several studies of fungal antigens causing systemic mycoses, such as coccidioidomycosis [169], paracoccidioidomycosis [43], cryptococcosis [158], histoplasmosis [128] and sporotrichosis [159]. Recently, immunoproteomic studies aimed at identifying exoantigens of the *Paracoccidioides* complex have been carried out [43]. In this study, a total of 15, 14, 33 and 17 exoantigens were identified for the *P. lutzii*, *P. americana*, *P. restrepiensis* and *P. brasiliensis* species, respectively. In addition, through bioinformatics analysis, 44 epitopes exclusive to this complex were predicted, with great potential for application in diagnostic tests exclusive to PCM.

In *Coccidioides* spp. antigens that are candidates for vaccines or to diagnose coccidioidomycosis were identified by Tarcha et al. [170]. Immunogenic molecules were detected in the cell wall of *C. posadasii*, among these aspartyl proteases was one of the highlighted molecules. During in vitro tests it showed the production of pro-inflammatory cytokines and the stimulation of T lymphocytes. During in vivo trials it led to increased survival of mice, resulting in a reduced fungal burden on the animals’ lungs. In other experiments with *C. posadasii* cell wall proteins, the enzymes aspartyl protease, phospholipase B and alpha-mannosidase showed epitopes that were recognized by MHC class II molecules. In addition, vaccination with a combination of the three epitopes provided protection for animals challenged with *Coccidioides* spp. [154]. Moreover, analyses of proteins such as glycosylated β-glucosidase 2 (GL2ur) [171] and the antigens F0–90 and F60–90 [169] presented the potential to be used in the diagnosis of coccidioidomycosis.

Immunoproteomic studies were also used to analyze antigens in *Cryptococcus* spp. For *C. gattii*, studies described the identification of 37 antigens, highlighting the immunodominant antigen 14-3-3 in all tests performed [165]. Analyses in the same segment were performed by Martins et al. [158], where antigens such as CG01, CG02, CG03 and R265 were identified by mass spectrometry. Through immunoinformatics, 374 peptides of B cells were characterized, making it possible to speculate that these molecules may be initial targets for the development of immunodiagnosis for cryptococcosis.

Additionally, immunoproteomic techniques have also been used in the study of systemic mycoses such as histoplasmosis, sporotrichosis and paracoccidioidomycosis. Studies carried out by Almeida et al. [128] showed the detection of 132 antigens of *Histoplasma* spp. Among these, molecules such as M antigen, P catalase and YPS-3 were mapped and presented 16 exclusive B cell epitopes, indicating that they are excellent molecules to be used in the use of new methods of diagnosis of histoplasmosis.

The immunoproteome of *Paracoccidioides* spp., performed by Moreira et al. showed the identification of 79 exoantigens of the *Paracoccidioides* complex [43]. Through bioinformatics tools, 44 exclusive epitopes of this complex were identified by bioinformatic tools, which can be excellent molecules to be used in the diagnosis of PCM. Using proteomic approaches, Rodrigues et al., [172], identified a total of 25 and 16 antigens from *P. brasiliensis* and *P. lutzii*, respectively. In addition, 29 proteins were characterized as new antigens of *Paracoccidioides*. Among the identified antigens, enolase, associated with the cell surface, was characterized as one of the main antigens of *P. lutzii* in human PCM [43].

Thus, the versatility that immunoproteomic techniques can provide is evident. Such techniques are extremely important for the identification and characterization of fungal antigens.

#### 3.1.3. Peptide Microarrays 

Identifying the antigens and epitopes involved in infectious diseases in detail is important for the understanding of immunopathogenesis, as well in enabling the development of vaccines and the knowledge of diagnostic and therapeutic targets. In this sense, the peptide microarray has been considered a revolutionary biotechnological approach, because it provides an extremely rapid and robust method which allows thousands of targets to be tested simultaneously. This technique comprises solid flat substrates (usually glass slides) with a collection of peptides, with known specificities, which are immobilized in discrete spatial locations [173]. With time, advances have provided the development of a high-density peptide microarray, where individual peptides are synthesized in situ on a glass slide at high densities [143].

The results found in this approach have constantly been expanding. They are promising and are increasingly being used in the biomedical area for the development/identification of new biomarkers, as well as for screening antigens, delivery systems and drug discovery and are mainly used in viral diagnosis and cancer research [174,175].

One of the peptide microarray applications is an identification of targets for vaccine candidates. A peptide microarray with 7466 unique peptides derived from 61 *Mycobacterium tuberculosis* proteins was constructed by Gaseitsiwe et al. The authors used sera from 35 healthy people and 34 from people with active pulmonary tuberculosis. With the data generated, the authors managed to combine epitopes of *M. tuberculosis* that bind to the main class II histocompatibility complex with peptide epitopes that were recognized exclusively by IgG from tuberculosis patients. Thus, this study makes clear the importance of peptide microarrays in identifying patterns of antibody reactivity, as well as in elucidating significant targets for the development of vaccines [144].

Another use for the peptide microarray technique was developed in the construction of a platform for bacterial binding assays. The authors describe the use of a random sequence microarray to identify peptides competing with the binding of bacteria to lipopolysaccharides. Through this study, of the 10,410 peptides studied, 54 demonstrated a high rate of inhibition of the bacterium–peptide interaction in a competition trial [176].

Fungal peptides, despite their potential antifungal activities, have seen their intracellular protein targets being poorly reported on. A study performed by Shah et al. (2019) used the yeast protein microarray approach to identify lactoferricin B (Lfcin B) and Histatin-5 yeast protein targets, due to their antimicrobial peptide mechanisms. A greater number of synthetic lethal pairs were found in Lfcin protein targets. Thus, these results demonstrated a greater lethal effect of Lfcin B in yeast [177].

It should be highlighted that the peptide microarray, while allowing up to hundreds of thousands to be tested, as in the case of high-density peptides, is not sufficient to cover complete proteomes. Furthermore, although this peptide approach is good at detecting linear epitopes, it is difficult to bind antibodies that need specific conformations and longer sequences [143]

#### 3.1.4. Cell Surface Shaving

The identification of new antigens for the development of vaccines as well as the development of serological tools to diagnose diseases is the objective of several studies. The shaving technique has been shown to be effective in identifying antigenic proteins exposed on the cell surface, which proves to be one of the best targets for host immunity [145]. 

In this approach, intact pathogens cells are incubated with enzymes, such as trypsin, for the hydrolysis of proteins and the released peptides are analyzed by high performance LC-MS/MS. This technique is performed mainly in Gram-positive organisms due to the thick peptidoglycan wall that resists digestion without lysis. In contrast, it is a technique less used in other microorganisms due to the thinner cell wall [145,146]. Several studies were performed to tackle this problem. In 2010, Hernáez et al. [178] established a fast and easy methodology, capable of identifying cell surface proteins in yeasts of *Candida albicans* through a non-gel proteomic approach based on a short period of trypsin treatment followed by peptide separation and identification using nano-LC followed by MS/MS. Recently, the shaving technique was used by Voltersen and collaborators [179] to identify a protein called conidial cell wall protein A (CcpA) as an important fungal spore protein involved in pathogenesis of aspergillosis. CcpA acts as a conidial stealth protein, altering the structure of the conidial surface to minimize innate immune recognition. This fact suggests that in the future it could be used as a possible immunotherapeutic or diagnostic molecular target.

#### 3.1.5. Phage Display

Several techniques aim to assist in the search for new molecular targets for the development of new drugs, diagnostic methods, and vaccines. Developed in 1985, with the advantages of simplicity, high efficiency and low-cost, the phage display technique uses recombinant proteins or peptides coupled to the surface of bacteriophages (phages), resulting in the expression of a heterologous protein on the surface of the viral capsid. Then, they can interact with different targets, allowing the selection of ligand–receptor pairs [147]. 

Research on systemic mycoses has benefited from this technique. For example, phage display was used to search for new therapeutic strategies for PCM. Oliveira et al. (2016) [180] used two phage display libraries to identify four peptides capable of inhibiting up to 64% of *Paracoccidioides* adhesion to pneumocytes in vitro and up to 57% of adhesion to extracellular matrix (ECM) components. Subsequently, these peptides were used to treat *Galleria mellonella* larvae before infection by *Paracoccidioides*. They demonstrated that the peptides increased the survival of *G. mellonella* infected by *P. brasiliensis* by up to 64% and by up to 60% in those infected by *P. lutzii*. In addition, Portes et al. (2017) sequenced the binding phages and through an immunoassay evaluated the interaction with positive and negative PCM sera. They observed satisfactory recognition (sensitivity of 74.19% and specificity of 71.43%) of a phage clone (LP15) in sera from patients with PCM, which can be useful for identifying new epitopes that can be applied in PCM serodiagnosis [181]. Another study, using a peptide screened for phage display, described a new system of drug distribution via oral administration aimed at the treatment of *C. neoformans*, able to increase the survival of mice in a model of infection [182].

### 3.2. In Silico Approaches for Antigen Prediction 

The choice of antigen remains the key component of the different immunodiagnostic tests [183]. The identification of antigenic/immunogenic regions in antigenic proteins is a key step for the diagnosis of infectious diseases and for antigen identification. In the past, the development of diagnostic tests and vaccines was based on the use of complete antigens and empiric methods. With the emergence of immunomics and immunoinformatics it is possible to map a specific antigen and to identify the epitope with great potential for targets in vaccine and diagnosis development [150]. Thus, bioinformatics strategies have a great advantage over the conventional methods in the development of diagnoses and vaccines for two special reasons: (i) in silico analyses are faster and more cost-effective, and (ii) the possibility of identifying proteins that are not expressed in vitro [184]. 

There are several successful examples of the application of immunoinformatics in the identification/development of new diagnostic targets as described with COVID-19 [185,186,187], *Schistosoma mansoni* [188], *Toxoplasma gondii* [189] *Fasciola hepatica* [190], helminth [191] human brucellosis [192] and *Leishmania* [193]. In the context of systemic mycosis, the number of works is substantially limited, and this approach has been used combined with immunoproteomic analyses as described to *Paracoccidioides* complex [43,172], *Sporothrix schenckii* complex [159] and *Cryptococcus gattii* [158].

In the following sections, we will discuss some approaches in bioinformatics that may improve the analysis in research with fungal pathogens, especially in the identification of new targets for vaccine and diagnostic assays. We list the most used tools, B-cell epitope and antigenicity prediction which are a key step in the identification of new candidates. Furthermore, tools for location and function prediction that assist in characterization of new targets are also described.

#### 3.2.1. B-Cell Epitope Prediction

The identification of B-cell epitopes is of great importance for medical applications as well as in terms of diseases’ control, diagnosis and vaccine development [194,195]. Thus, the prediction of these epitopes is essential in the context of the modern analysis and development of vaccines and diagnostics. However, B-cell epitope mapping is the corner-stone step in the production of diagnostics, while [160] it is only the first step in designing potent vaccines. In addition, B-cell epitopes can be linear (continuous) and conformational (discontinuous). 

Linear epitopes have their amino acid residues organized in the primary sequence of the protein, while discontinuous epitopes are formed by residues organized far apart in the primary structure, but which come nearer because of protein folding [196]. Linear epitopes represent only 10% of B-cell epitopes and normally the amino acid sequence is required for prediction. On the other hand, conformational epitopes represent 90% of the total B-cell epitopes, but demonstrate the difficulty of prediction in cases of neglected tropical disease because they frequently require the PDB format as input [196], with a few exceptions [197]. 

Currently, several approaches have been proposed in linear and discontinuous B-cell epitope prediction. For linear epitopes, it is possible to highlight algorithms such as BcePred [198], BepiPred [199], ABCpred [200], COBEpro [201], BCPREDS [202], SVMTriP [203], LBtope [204] LBEEP [205], IEDB [206] and BEST [207]. Among them, BCPREDS, ABCpred, BepiPred, SVMTriP and CoBepro were used in the immunogenic and diagnostic studies against *C. albicans* [208], *A. flavus* [209] *S. schenckii* complex [159] *C. gattii* [158] and *Paracoccidioides* spp. [43,172]. The method of prediction of these tools consists of the combination of multiple physico-chemical properties (such as BepiPred), while other tools associate physico-chemical properties with learning machine models, such as the Hidden Markov Model (HMM), Support Vector Machines (SVMs) and Neural Networks, which improve the efficiency of the analyses significantly [210]. The ABCpred is based on an artificial neural network and trained with epitopes from viruses, bacteria, parasites, and fungi with an accuracy of 65.93% [200]. SVMTriP (2012) predicts linear epitopes using Support Vector Machines with a sensitivity of 80.1% and a precision of 55.2% [203]. BCPREDS uses a kernel method for predicting epitopes, achieving a predictive performance of AUC = 0.758. Furthermore, the current implementation of BCPREDS allows the user to select from three prediction methods: (I) AAP [211]; (II) BCPred and (III) FBCPred in the same tool [203].

In the context of tools for the prediction of discontinuous epitopes, BEST, BepiPred-2.0 and CBTOPE have prominence because they realize the prediction by using primary sequence proteins. On the other hand, tools like DicoTope [212], SEPPA [213], BEpro server (formerly known as PEPITO) [214], Ellipro [215] and EPITOPIA [216] use structure-based approaches and require 3D structure information. There is little comparative data analyzing the performance of these tools. However, EPITOPIA yields a higher success rate of 89.4% when compared to ElliPro, used in the analysis of immunogenic properties of the biopharmaceutical enzyme uricase from *Aspergillus flavus, Bacillus subtilis* [209] and DiscoTope [217] and, in another study, SEPPA gave the best performance among the six tools, followed by DiscoTope and BEpro [218]. Nevertheless, similar to tools for linear epitope prediction, it is difficult to appoint the best tool, but it is recommended to use different methods for discontinuous epitope prediction [219].

Lastly, there is a single database of immunoinformatics specifically for fungi. FungalRV [220] (fungalrv.igib.res.in) is a server which has gathered several tools, including adhesin predictor, cellular localization, linear and conformational B-cell epitope predictor, and T cell epitope predictor. One detailed protocol of FungalRV can be found in the work of Chaudhuri and Ramachandran [221]. 

Thus, we propose in Figure 1 a simple and efficient workflow to predict linear and discontinuous B-Cell epitopes.

#### 3.2.2. Antigenicity Prediction 

The use of antigenicity prediction is an important step during the identification of targets for diagnosis. Currently, it is possible to cite several tools for this analysis, such as Vaxijen [222], NERVE [223], Vaxign [224], ANTIGENpro [225], Jenner-Predict server [226], iVAX [227] and VACSol [228]. However, only Vaxijen v.20 [229] was trained with fungal data and because of this it is indicated for these pathogens. Its prediction uses FASTA sequence and is independent of alignment with antigens confirmed experimentally but is based on physico-chemical properties of proteins with an accuracy between 70% and 89% [222]. This tool was developed in 2007, it is available on an online server and is one the most cited algorithms when regarding antigenicity [230] (Figure 1).

#### 3.2.3. Location Prediction

The prediction of location and function are additional steps in the identification of new targets. Protein location is especially important in the context of vaccine and diagnostic targeting, for the simple fact that secreted or extracellular proteins are more likely to interact with the host’s immune system. 

In the context of fungal analysis, the tools most applied are SignalP (SP prediction and transmembrane domain) [231,232,233,234,235], followed by Phobius (SP prediction and transmembrane domain) [236,237], TMHMM (transmembrane propellers) [238] and TargetP (subcellular location: chloroplast, mitochondria and extracellular) [239]. For multiple subcellular locations prediction, WolF Psort [240] based on identification of SP and TM signals, amino acid composition and functional motifs, such as DNA binding motifs. WolF Psort is the most cited, with the broadest set of fungal proteins, and has been appointed as the best individual tool for fungal data analysis [241]. Another alternative, MultiLoc2, which uses the composition of amino acids, the presence of known classification signals, phylogenetic profiles, and terms of GO in its prediction, presents a better performance than WolF Psort in both animals and plants [242]. There are still two specific tools for protein subcellular localization: ProtComp and BUSCA, which combine prediction based on homology analysis, structural properties of proteins and prediction of certain functional peptide sequences (SP, GPI anchors, mitochondria transit peptides and transmembrane segments). These tools are user friendly and accept the input sequence in FASTA format. They are all available to use for free online or in a downloaded version.

However, several studies have identified proteins in the extracellular environment that do not present signal peptides, suggesting that there are alternative secretion pathways. These secretory pathways that act without the involvement of an N-terminal signal peptide are classified as unconventional or non-classical secretory pathways [243,244], which are difficult to predict due to the diversity of mechanisms involved in the secretion process, generating great difficulty in the construction of computational tools [234]. This limitation mainly affects analysis of fungal proteins due to the predominance of the number of proteins that make use of these alternative pathways, as seen in work with *P. brasiliensis* [121,245], *H. capsulatum* [246], *C. neoformans* [247], *C. albicans* [248,249], *A. fumigatus* [250] and *S. cerevisiae* [251]. 

Currently, there are three tools available to perform non-classical secretory pathway analysis: SecretomeP [252], OutCyte [253] and SPRED [254]. All of them make use of classic secretory proteins based on the hypothesis that all secretory proteins share common features regardless of the specific pathways. Unfortunately, SPRED is not available online and the comparative analysis, SecretomeP performs the best in the identification of proteins of the non-classical secretory pathway when compared with OutCyte [253]. 

In Figure 2 we propose a workflow for predicting cell localization of fungal proteins. 

#### 3.2.4. Functional Characterization of Protein

The prediction of cellular location is directly associated with functional protein characterization. These processes include a significant assignment of bioinformatics to comprehend the disease’s mechanisms, diagnosis development, drug targets and vaccines discoveries. 

In fact, most proteins deposited in databases are described as “hypothetical proteins” —with the prediction based on open reading frame (ORF)—or “conserved hypothetical proteins” where proteins are present in several phylogenetically related strains, but without functional validation [255]. This large proportion of unannotated proteins is due to both the specificity of the protein in question, as well as the absence of manual analysis. For example, in the construction of ParaDB, through the re-annotation by in silico analysis, it was possible to reduce the rate of unannotated proteins in the genus *Paracoccidioides* from 60–90% to 25–28% [256].

It is particularly important to highlight the potential role of hypothetical proteins in the infection process [257] and in vaccine development [258], as well as their role as diagnosis markers [43], drug targets [259], and in understanding physiological and biochemical pathways [257,260,261]. The analysis of these proteins is frequently neglected because of the absence of functional information. This situation demonstrates the need to comprehend/improve the functional characterization processes of these proteins. In this context, we propose, as supported by the literature [255,257,258,262,263,264,265,266], one in silico workflow for functional prediction of hypothetical proteins (Figure 3). 

The first step is the search for homologous proteins in databases using BLAST and PSI-BLAST tools. If a homologous protein is identified, it is possible to search for GO terms using the tools mentioned above. If no homologous proteins are identified, we proceed to predict location (Figure 3), with analysis of identification of functional domains done by InterProScan, followed by prediction of the tertiary structure by Swiss-model or I-TASSER, both are free webservers that perform the prediction based on the primary structure of proteins. With the 3D protein structure, it is possible to search in the database for a similar model, which may help in comparing the region of the active site or binding site, for example. Lastly, it is also possible to find a functional protein association by the prediction of protein–protein interactions using, for example, STRING (https://string-db.org/, accessed on 20 February 2022) based on the principle that proteins often interact with one another in a mutually dependent way to perform a common function [267,268].

## 4. Strategies to Improve the Diagnosis of Human Systemic Mycoses Using the Available Technological Approaches

As described above, it is remarkably difficult to select the best diagnostic strategy for each systemic mycosis. Early diagnosis is known to be pivotal in order to treat effectively. In this sense, we suggest approaches to improve the diagnosis of the main human systemic mycoses, which were discussed in this review, according to the current challenges. 

In the PCM context, current challenges regarding its diagnosis include: (i) increasing the sensitivity and specificity, reducing the presence of cross-reactivity and false-negative results on serology, mainly by the detection of serum antibodies; (ii) develop and standardize a molecular approach to identify phylogenetic species of the *Paracoccidioides* genus; (iii) suggest a cure control test able to evaluate the recovery of the cellular immune response and, (iv) develop tests that are capable of running fewer samples, since, generally, the demand for it is low. 

Considering epitope prediction tools, experimental approaches such as immunoproteomics and phage display could be excellent strategies for the identification of antigenic targets that are more sensitive and specific for serological tests. For the development of a highly sensitive molecular test, such as TaqMan Multiplex Real-Time PCR, which potentially is able to identify and genotype *Paracoccidioides* species directly from clinical samples (such as sputum, tissue, bronchoalveolar lavage (BAL) material, cerebrospinal fluid (CSF), wound injury and peripheral blood). This would also be a great option for routine diagnosis. In this specific case, the LAMP has been used in sputa samples [269] and formalin-fixed paraffin-embedded (FFPE) tissue blocks [270] from humans and armadillos. LAMP has the advantage that it is fast, inexpensive, sensitive, without cross-reactivity, and false-positive results do not occur because it uses the detection of the single nucleotide sequence for *P. brasiliensis* complex, unlike some immunological tests that use indirect detection methods for microorganisms. However, LAMP assays, described above, were designed without considering *P. lutzii* as a biological species. Following that, a new set of primers for use in the LAMP technique were developed to differentiate *Paracoccidioides* species [271]. 

The search options for a good marker of re-establishment of the cellular immune response, which is important to establish the end of antifungal therapy, could involve the T-Cell epitope prediction approach. A lab-based approach for this immunological assessment has been recently published [272]. Finally, we propose the lateral flow assay (LFA), which has been used for *Histoplasma* antigen testing [273] and requires minimal laboratory equipment and infrastructure. Another option is the Dot-ELISA assay for *Paracoccidioides* species antibody detection, as suggested for *P. brasiliensis* serologic screening [42]. Indeed, these applications could be extended to all *Paracoccidioides* species, using new antigenic targets defined by in silico prediction and/or immunoproteomics.

Antigen detection assays are usually more effective than antibody testing for histoplasmosis diagnosis and are particularly useful in acute disease, mainly in immunocompromised individuals, who frequently have the disseminated form of histoplasmosis without detectable antibodies for the fungus. 

The detection of *Histoplasma* galactomannan antigens is an important approach in its diagnosis. However, these tests are not universally available [274]. Therefore, the main challenges in histoplasmosis are as follows: (i) new options of antigenic targets to *Histoplasma* antigen detection tests; (ii) the improvement of molecular diagnosis. First, the development of different polyclonal or monoclonal antibodies that can identify antigenic targets with high specificity and sensitivity, and which can be widely available, mainly in low-income countries, would be a key goal. In this sense, several approaches described in this review could be used to look for antigenic molecules for development of these antibodies, such as immunoproteomics, phage display and antigenicity prediction. Second, the standardization of a cost-effective molecular approach that is sensitive enough to detect a low fungal burden, mainly in whole blood, would be useful. Thus, the search for multi-copy genes or repetitive sequence markers in the *Histoplasma* genome might be considered in the search for new DNA fungal targets.

Currently, the identification of *Aspergillus* at the species level is one important challenge. It is relevant because, recently, the presence of ‘cryptic’ *Aspergillus* species has been revealed in clinical samples from patients. In addition, an increase in the profiles of antifungal resistance to voriconazole, itraconazole and caspofungin has been considered as the therapy of choice for invasive aspergillosis. On the other hand, more recently, COVID-19 associated pulmonary aspergillosis (CAPA) has been reported with concern [275,276]. Studies indicate that, if actively investigated, approximately 20% of severely ill patients with COVID-19 would have an invasive aspergillosis diagnosis [277]. The point is that few clinical laboratories do molecular assays or expensive serology-based tests such as galactomannan and β-D-glucan. In addition, due to hazards related to aerosol production, bronchoscopic or non-bronchoscopic lavage procedures are rarely used. In this context, standardization of a breakpoint for the GM assay in other biological fluids, such as tracheal aspirate, needs to be validated to help the diagnosis of CAPA. 

Currently, the diagnosis of cryptococcosis is the most well established among the described human systemic mycoses. However, the species identification remains relevant for therapy since *C. gattii* infections tend to be more resistant to treatment. Therefore, we highlighted the need of a proposal for genotyping of *C. neoformans* and *C. gattii*.

## 5. Conclusions

It is pivotal to improve the approaches used to identify new candidates for the diagnosis of systemic mycoses. It is not only about diagnosis, but also epidemiological perspectives, new targets for patient follow-up, as well as therapeutic adjuvants. We emphasize the need for incorporating simpler and more sensitive molecular approaches in diagnosing mycoses. Finally, the development of fast and inexpensive tests that can be used at point-of-care (POCT or bedside testing) is essential, as suggested [278]. These improvements could reduce the time needed to establish the diagnosis, as well as positively influence patient outcomes and follow-up, resulting in less mortality, morbidity and sequelae.

## Figures and Tables

**Figure 1 pathogens-11-00569-f001:**
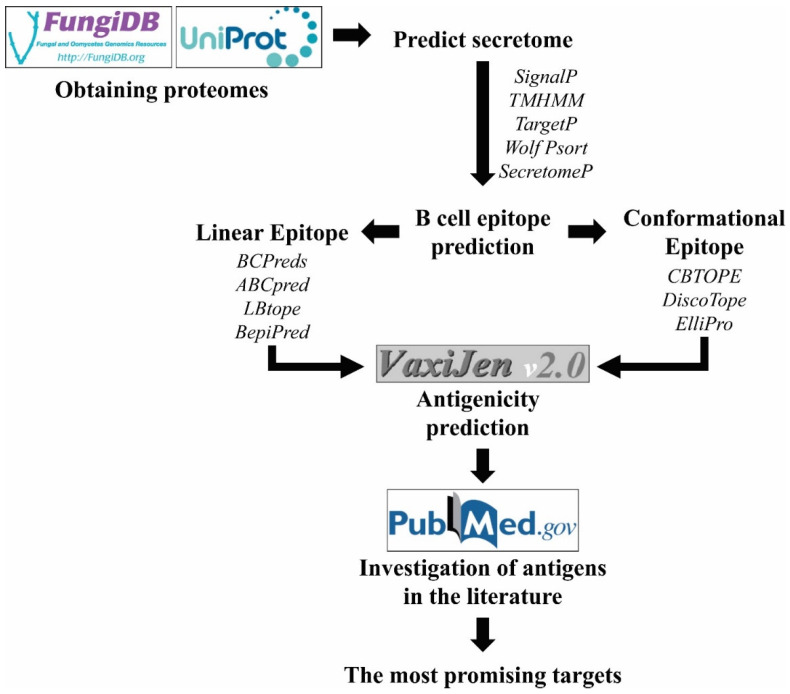
Workflow of epitope prediction for diagnostic development. Obtaining fungal protein or proteomes (FungiDB). Secretome prediction by SignalP, THMHMM, TargetP, Worlf Psort and SecretomeP. The B-cell epitope prediction—Linear Epitope: BCPreds, ABcpred, LBtope and BepiPred; Conformational Epitope: CBTOPE, DiscoTope and ElliPro. The antigenicity evaluation by VaxiJen. Investigation of new target information in literature.

**Figure 2 pathogens-11-00569-f002:**
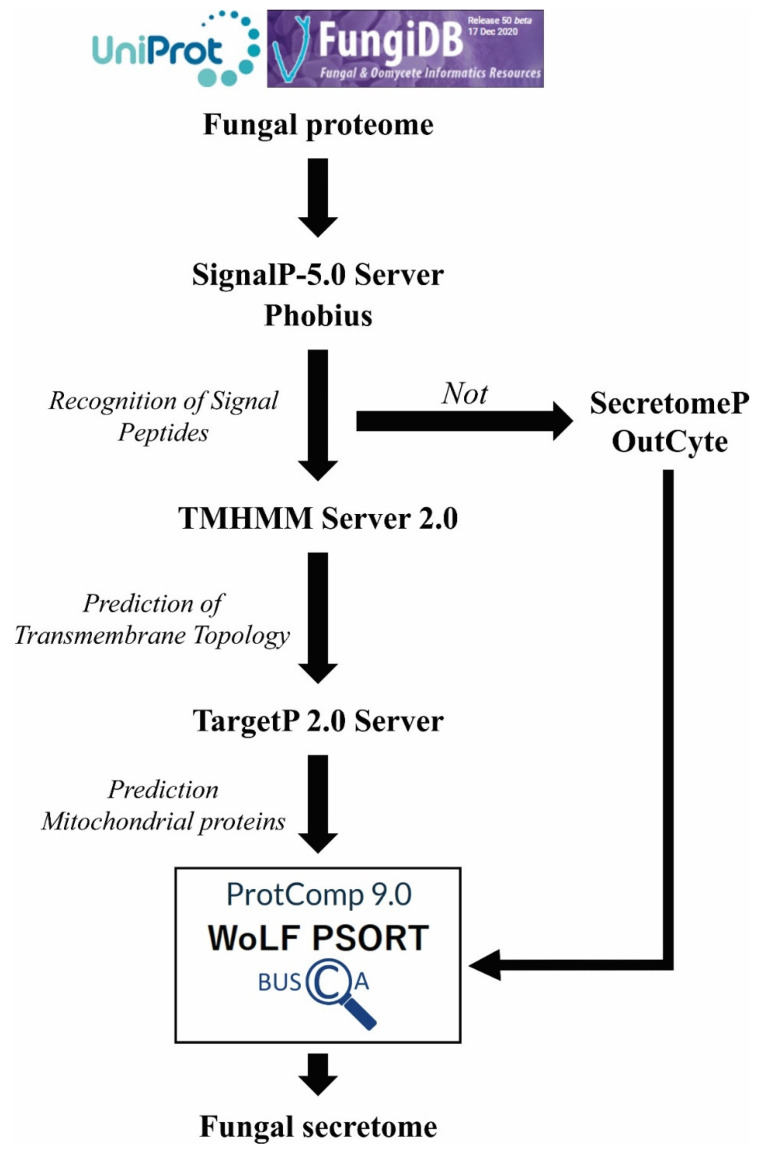
Workflow of Location Prediction. UniProt and FungiDB for protein sequence obtention; SignalP/Phobius (for signal peptide prediction), TMHMM, (transmembrane helices), TargetP (mitochondria), ProtComp9.0/WoLF PSORT/BUSCA (cell localization) and SecretomeP/OutCyte (non-classical secretion) and FragAnchor and PredGPI (GPI anchor).

**Figure 3 pathogens-11-00569-f003:**
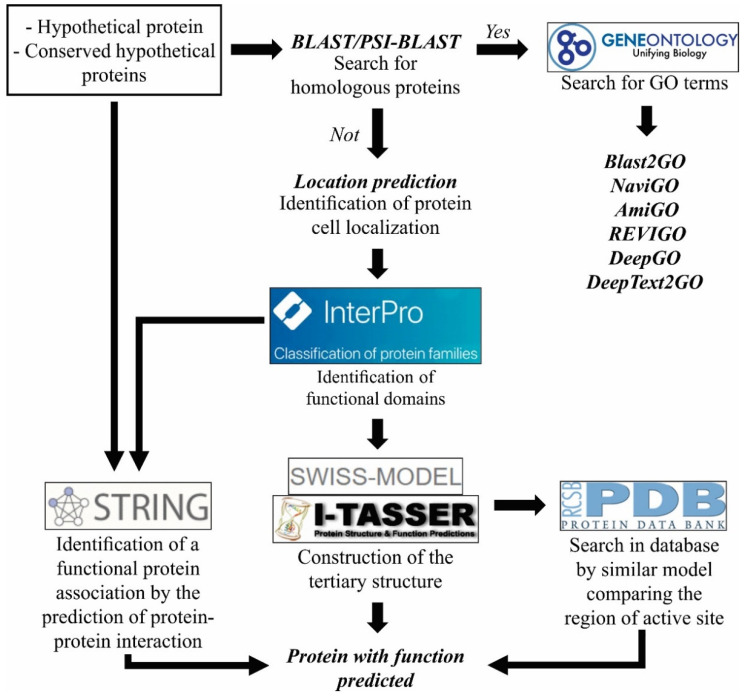
Workflow of functional characterization of hypothetical proteins. Searching for homologous proteins by Blast and PSI-Blast. Investigation of GO in homologous sequence by Blast2GO, NaviGO, AmiGO, REVIGO, DeepGO and DeepTex2GO for homologous proteins in database. For non-homologous proteins in database: Location prediction; Searching functional domains on InterPro; Construction of the 3D model protein by I-TASSER/SWISS-MODEL and comparing protein models on PDB and/or investigation of protein–protein interaction by STRING.

## Data Availability

Not applicable.

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
