# Peer review of "Challenges in Serologic Diagnostics of Neglected Human Systemic Mycoses: An Overview on Characterization of New Targets"

_pathogens, 2022, doi:10.3390/pathogens11050569_

Round 1

Reviewer 1 Report

Manuscript Title: Challenges in serologic diagnostic of human systemic mycoses: An overview on characterization of new targets.

Journal: Pathogens

Brief Summary: The authors provided a very detailed review article about the current and experimental diagnostic approaches for five human fungal pathogens; Paracoccidioides, Histoplasma, Aspergillus, Coccidioides, and Cryptococcus. They provided a concise description of the fungal organisms and their primary manifestation of human disease. They concisely provided background regarding the importance and complexity of the measures of a diagnostic test (i.e. sensitivity and specificity). They briefly reviewed and discussed the current diagnostics for each of the fungal organisms. Finally, they focused heavily on thoroughly describing the approaches used to search for new diagnostic candidates. An updated review of diagnostic tests for fungal organisms is much needed and I believe this manuscript fills this need. This manuscript is written well and easily understood overall but does require some minor revisions.

Comments: This is an outstanding manuscript but does require some minor changes and additions regarding figures, tables, and discussion.

First, Figure 1 is unclear in its meaning. I suggest Figure 1 be completely re-designed. In addition, for the updated Figure 1, it is important to include a description to help guide the reader through this figure (Line 52). There are two figures labeled as Figure 1 (line 51 – 52 and line 744 – 749). This needs to be changed and make sure that the manuscript text reflects this change. The rest of the figures following must also be adjusting.

Second, Section 2 of the manuscript would greatly benefit from a concise table that, for each of the fungal pathogens, details the types of diagnostics tests, the source of the human specimen that is used in this test, sensitivity/specificity measures, time to complete diagnostic test, main advantages/disadvantages of the tests and if such diagnostic tests are inaccessible due to expense, infrastructural resources, or personnel expertise. Such a table is drastically needed to aid comprehension of Section 2 of the paper.

Third, Section 3 would greatly benefit from a summary table that details each of the approaches along with their sensitivity/specificity, the associated fungal organism, time to result, advantages/disadvantages, infrastructure requirements, etc. For each of the new diagnostic candidates there needs to be a concise discussion about the economic/infrastructural requirements and advantages they offer over the current standard diagnostic tests. In addition, for these new diagnostic candidates, please discuss the feasibility of use in the areas where they are most needed. For example, how feasible is it for a rural community hospital with unstable electricity and limited technology to adopt and use these new diagnostic candidates? This type of discussion is included in lines 883-892, however I suggest each of the new diagnostic candidates also have a sentence discussing feasibility of use and the infrastructure requirements.  

Overall, this is an excellent manuscript that only requires the modification of figures and addition of two tables to improve the comprehension of this extremely thorough review. 

Author Response

test

Reviewer 2 Report

The manuscript presented for review is well-written report about the challenges in serologic diagnostic of human systemic mycoses and alternative approaches for the identification of new targets for fungal pathogens, which can help in the development of new diagnostic tests. Mycoses are commonly chronic diseases that are difficult to diagnose and treat especially among immunocompromised patients. Therefore, the subject of the review is important and significant.

The authors divided the work into four chapters. In the first of them they presented all fungi species which caused mycoses, then describe the diagnosis of mycoses. The next chapters describe the search for new therapeutic options and strategies to improve the diagnosis using the available technological approaches. Data are presented in a very exhaustive way which makes the review very long. It needs to be shortened and rearranged. Moreover, in order to present the data in a clearer way, I suggest adding figures or tables, especially to chapters 3 and 4. It will also allow to summarize the information contained therein.

Summarizing, the review is interesting and well structured, but it is necessary to do a major revision to be accepted.

Specific suggestion/comments:

Literature references should be placed in square brackets.

Line 8: please remove the underline

Line 47:  diagnostic tests à diagnostic tests.

Line 61: in order to à to

Line 89: Europe à Europe,

Liner 92: subacute à subacute,

Line 99: South à South,

Line 110: C. posadasii à Coccidioides posadassi

Line 116: Argentina à Argentina,

Line 125: actually both à both

Line 133: the majority of à most of

Line 180: in vitro à in vitro

Line 253: HIV/AIDS à the sentence should be reworded to indicate that it is people infected with HIV and suffering from AIDS

Line 225: et al à et al.

Line 313: Coccidioides immitis à C. immitis

Line 337: test, and à test and

Line 361: sensitivity and à sensitivity, and

Line 433: <100 cell / mm3 à <100 cell/mm3

Line 434: <200 cell / mm3 à <200 cell/mm3

Line 471: In other works à In other works,

Line 475: in order to à to

Line 475: in vitro à in vitro

Line 541: in vitro à in vitro

Line 605: in order to à to

Line 625: et al à et al.

Line 637: diagnostic methods and vaccines à diagnostic methods, and vaccines

Line 691: as a consequence of à because of

Line 871: (337), and à (337) and

Line 884: are able to à can

Line 908: therapy, since à therapy since

Line 910: Future perspectives and concluding remarks à Conclusions

References should be changed according to MDPI requirements, e.g.:

  1. Bongomin, F.; Gago, S.; Oladele, R. O.; Denning, D. W. Global and Multi-National Prevalence of Fungal Diseases—Estimate Precision. Fungi 2017, 3 (4). https://doi.org/10.3390/jof3040057.

Author Response

Please, find the attached file with our responses.

Thank you 

Best regards

Reviewer 3 Report

The paper focuses on some important systemic mycoses, although NOT the numerically most important(i.e: systemic candidiasis is ommited). Whereas histoplasmosis, coccidioidomycosis or paracoccidioidomycosis may represent an important burden for endemic countries, their impact in other parts of the world is limited.

Whereas most of the mycoses of interest cited in the paper are characterized by pulmonary involvement, it is not true that most of them commonly affect the skin. Moreover, dissemination is usually restricted to patients with a severely hampered immune system.

The title suggest the reader that the review will discuss the present and future direction of serodiagnosis. The content of the fist half of the paper, however, wanders around the different diagnostic techniques available only for each of the 5 mycosis under the authors’ interest. Serodiagnosis includes the detection of antigen and antibodies and it would have been desirable to find in the text a comprehensive description of currently available techniques for THESE specific purposes in an ordered fashion, along with a discussion about their strengths and weaknesses.

For instance: Antigen detection of Aspergillus by immunochromatography skips one of the two currently available tests and does not discuss the differences in diagnostic yield as compared to each other and barely compares them to the classic EIA technique. No mention can be found to the detection of antibodies for the diagnosis of other Aspergillus related pathologies such as chronic pulmonary aspergillosis or bronchopulmonary allergic aspergillosis. In contrast, big strenght is put on the antibody detection of coccidioides skipping the development of new antigen detection test, that may be more useful for the early detection of an acute infection.

The second part, in contrast, is an interesting summary of potential future options under development.

My suggestion is to reorder the first half of the paper: to shorten the description of non-serological techniques and to fulfill the gaps related to currently available techniques that are not mentioned in the paper, as well as to eliminate biased messages that do not reflect the true situation of the fungal diagnosis in the real world.

Author Response

(The authors gave the same response as above.)

Round 2

Reviewer 2 Report

I would like to thank the Authors for thoroughly addressing the review comments.

Author Response

Thank you for your feedback and suggestions. The suggestions greatly improved the quality of the article.

Best regards

Clayton

Reviewer 3 Report

Not sure whether Aspergillosis or Cryptococcosis can be considered “neglected mycosis”. 

In general: As the paper is focused on serodiagnosis, it would be better to center the first part of the paper on the role of current serodiagnostic methods and their limitations. This has been done in part by deleting most on the information about molecular methods, improving the reading of the text. This comment applies particularly to the Cryptococcosis section. The paper is still rather long and the true novelty is found from section 3 onwards. Sections 1 and 2 add little to the paper, they  still have some innacurate statements, and what may be the most important related to the aim of the paper: do not highlight the drawbacks of currently marketed serodiagnostic methods. Sections 1 and 2, in fact can be considerably shortened.

Some mor specific comments to the paper in its current form are stated below.

Line 70: …some important human systemic mycosis

Line 100: amend “largest”. Incorrect word.

Lines 112-13: Incorrect distribution map description. As a genus, Aspergillus is widely found outside the Arctic and the Antarctic (https://www.discoverlife.org/mp/20q?search=Aspergillus)

Line 113: Aspergillus-related pathologies only occurs in hosts presenting predisposing underlying conditions. In this paragraph, the term “aspergillus-related pathologies” is more appropriate than “Aspergillus infections”

Lines 123-135: As it is also an endemic mycosis, coccidioidomycosis would be better placed after Histoplasmosis, and before Aspergillosis-Cryptococcosis.

Lines 232-234: The sentence does no make much sense.

Lines 359-361: by visualization of the yeast forms in clinical samples with the help of fungus-enhancing stainings, such as calcofluor white, or using histopathology techniques.

Lines 361-362: lack sensitivity, are time consuming, and require a degree of expertise to recognize the fungus.

Lines 426-427: As stated in the previous review, IMMY is not the only manufacturer with a marketed lateral flow diagnostic system. Moreover, the context in with lateral flow devices have been developed, in fact, is to give a rapid diagnostic response (point-of-care like).

Lines 429-431: Innacurate statement. Insert reference justifying it or delete the sentence.

Lines 433-35: Please, rearrange the message by putting together the information about GM. Description of molecular techniques has been removed from other pathologies, it is not easy to understand why this sentence is here.

Line 438: Anti Aspergillus IgG is not an anti-galactomannan antibody (10.1016/j.biotechadv.2018.03.016). An lateral flow technique for the rapid detection of these antibodies has been marketed in recent years.

Line 580 onwards: please substitute “tape” by “strip”.

Lines 635-656: Hard to understand why “3.1.1 Classical methods for serologic diagnostic” is under the heading “3.1. Experimental approaches”, and not before “3. Approaches used to search for new diagnostic candidates”. Moreover, most of the information herein is repeated from the first part of the paper.

Author Response

Please, find the attached file.

Best regards

Clayton

Round 3

Reviewer 3 Report

Minor amendments needed:

Line 281: replace anti-galactomannan by anti-aspergillus

Line 282: insert comment on the description of precipitins ".., although lacking specificity, ..."

Line 287: Wrong statement. ABPA is proportionally much more frequent in Cystic Fibrosis than in Asthma patients.

Line 330-31: Insert nuance “…provided commercial standard database are properly enriched,…”

Line 355: Amend “…for children, data on the accuracy of the test are limited…”

Table 1: Latex agglutination and EIA-based antigen detection tests not stated. Recommended to delete galactomannan detection by agglutination test as this is obsolete.

Author Response

Please, find the attached file

This manuscript is a resubmission of an earlier submission. The following is a list of the peer review reports and author responses from that submission.

Round 1

Reviewer 1 Report

This review starts as a review of current diagnostics for endemic fungi and then meanders into methods that will probably never be developed.  The authors would be better served by introducing molecular methods being developed such as real-time PCR and metagenomics which are more likely to be developed into viable assays in the near future.

Line 71:  Although they have been published, according to Index Fungorum and MycoBank, these species are not scientifically recognized.  You can either put ‘proposed species’, or stick to the two recognized species of Paracoccidioides.

Line 97:  Aspergillus is definitely not distributed “mainly in developing countries”.  This line should be removed.

Lines 122-124:  These are vague generalizations that have been proven to not be true.  C. neoformans and C. gattii SC occur in both immunocompromised and immunocompetent individuals.  The difference is that C. neoformans is more abundant so you see it more with HIV.

Lines 172-175:  MALDI is great for identification, but it is not considered a diagnostic test the way the other tests are.  With MALDI you already have an isolate and therefore a presumptive identification.

Line 173 and throughout:  Make sure all genus and species names are in italics.

Line 209:  Cross reaction to what?

Line 237-239:  It is not clear what you are saying here, it does not make sense in English.  Do you want to say antigens from different species can be pooled to overcome specificity?

Lines 262-262:  In the US, antigen tests are used almost exclusively, antibody tests are rarely used.

Line 279:  IMMY diagnostics has developed an antigen assay lateral flow device.  This should be mentioned here.

Line 304:  There is an LFA test for galactomannan that is available.  It should be mentioned here.

Lines 378-388:  Why are you including histopathological ID here but not for Coccidioides where it would be much more definitive?

Line 446-521:  This entire section can be deleted.  The fact that all these exoantigens have been detected, some over 30 years ago, and none of them have been used to develop diagnostics means that they are not going to be used to develop diagnostics!!

Line 606:  Vaccine are not diagnostics.  Research for vaccine candidates is not considered to be characterization for diagnosis.  The review is getting way off track here and all the data and references to vaccine development needs to be left out or the original scope of the manuscript needs to be changed. 

Line 691:  Again, this is not diagnosis, this is treatment.

705-744:  It is not clear how describing bioinformatics tools has anything to do with fungal diagnostics in that it is not put into context, the tools have not been referenced as being useful for any diagnostic purpose to date for fungi.  No reference is given as to how they are currently being applied to fungal diagnosis.

Line 788-851:  This has gotten way off track.  I can develop a diagnostic without eve having any idea of a protein’s function.  This is so far away from future diagnostic tools.

Lines 852-926: Same as above.

Reviewer 2 Report

  1. This manuscript needs to be rewritten and refocused, with a new title. It is only about new antigenic targets as diagnostic markers. It excludes an enormous amount of information on other new targets using other methods, such as next gen sequencing PCR, etc. It would probably be best to focus only on new targets that are antigenic or immunologic in nature. There are voluminous reviews on molecular assays that would be redundant in this review.

  1. There are some references throughout the manuscript to other diagnostic methods such as LAMP (a PCR method). This method is not new. If non antigenic methods are to be discussed, then the discussion needs to be restricted to new methods. Because of the large amount of information already in the manuscript dealing with antigens, it is probably best to delete other method discussion such as PCR, MALDI-TOF, etc.

  1. Figure 1. PCM needs to be spelled out to match the other fungal mycoses.

  1. Delete section 1 on human systemic mycoses. It is not related to the manuscript topic.

  1. Section 2 needs to be completely rewritten. It is a generalization that provides little information. It would be a good idea to discuss strengths and weaknesses of current antigenic or immunologic assays and what needs to improve—hence, the necessity of “new” target discovery.

  1. A table or tables of currently approved (US FDA or European CE mark) diagnostic tests that are antigenic or immunologic in nature would be very helpful and would condense the manuscript.

  1. In the section of peptide microarrays, there is a statement that, “Cell surface shaving is a useful technique for the identification of druggable targets and vaccine candidates”. While possibly true, the statement has no bearing on diagnosis and should be deleted along with many other sentences/paragraphs that meander into a discussion of vaccines.

  1. A table or a thorough discussion of companies that are developing their technologies into diagnostic tests would be very informative. Otherwise, the manuscript is mainly about research and not actually diagnostic tests that are in the pipeline and on their way to the diagnostic market.

  1. Section 3.2.1, Location prediction, does not mention any significant diagnostic applications. It is multiple paragraphs describing software. Are there any successful diagnostic papers using this method? Are any tests being developed? The authors should stay away from descriptions of research methods that have little chance or application to fungal diagnosis. They are distracting. Same for section 3.2.2.

  1. Section 3.2.4, Functional characterization of protein, makes no mention of how this is applied to development of a diagnostic test. It should be deleted. The same for the following section (3.2.5).

  1. In the abstract, the authors state, “Rapid, low-cost, simple, highly-specific and sensitive diagnostic tests are critical components of patient care”. They should explain how their “new targets” would fit into this claim.

  1. Although there are many different views of which fungi are important in systemic mycosis, in reality, the vast majority of fungal infections and most of the assays, such as those for sepsis, are targeted towards Candida. It would probably be better to also include Candida in the list of systemic mycotic agents and new targets.

  1. Section 3.1.1-2 has multiple paragraphs discussing Sporothrix, but in the list of major systemic mycotic agents, it was not included. Therefore it should be deleted.